# LIMO: Less is More for Reasoning

Yixin Ye[1,2]*, Zhen Huang[2,3]*, Yang Xiao[4], Ethan Chern[1,2], Shijie Xia[1,2], Pengfei Liu[1,2†]
[1]Shanghai Jiao Tong University  [2]SII-GAIR
[3]Fudan University  [4]The Hong Kong Polytechnic University

## Abstract

We challenge the prevailing assumption that complex reasoning in large language models (LLMs) necessitates massive training data. We demonstrate that sophisticated mathematical reasoning can emerge with only a few examples. Specifically, through simple supervised fine-tuning, our model, LIMO, achieves 63.3% accuracy on AIME24 and 95.6% on MATH500, surpassing previous fine-tuned models (6.5% on AIME24, 59.2% on MATH500) while using only 1% of the training data required by prior approaches. Furthermore, LIMO exhibits strong out-of-distribution generalization, achieving a 45.8% absolute improvement across diverse benchmarks, outperforming models trained on $100\times$ more data. Synthesizing these findings, we propose the Less-Is-More Reasoning Hypothesis (LIMO Hypothesis): In foundation models where domain knowledge has been comprehensively encoded during pre-training, sophisticated reasoning can emerge through minimal but strategically designed demonstrations of cognitive processes. This hypothesis suggests that the threshold for eliciting complex reasoning is not dictated by task complexity but rather by two key factors: (1) the completeness of the model's pre-trained knowledge base and (2) the effectiveness of post-training examples in serving as "cognitive templates" that guide reasoning.[1]

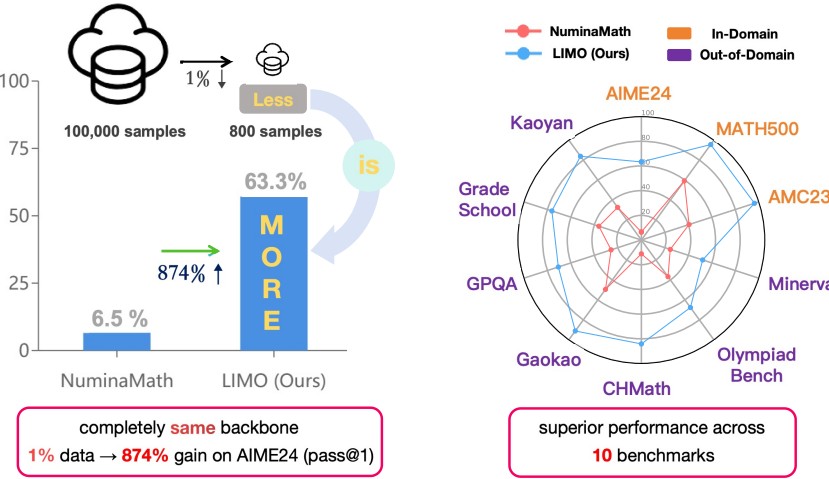

Figure 1: LIMO achieves substantial improvement over NuminaMath with fewer samples while excelling across diverse mathematical and multi-discipline benchmarks.

---

*Co-first authors

†Corresponding author

[1] https://github.com/GAIR-NLP/LIMO

# 1 Introduction

Complex reasoning has long been considered one of the most challenging capabilities to instill in large language models (LLMs). While recent work has shown that LLMs can be effectively aligned with user preferences through relatively small amounts of instruction data (Zhou et al., 2023), teaching models to reason—particularly in mathematics and programming—is widely believed to require vastly more training examples (Paster et al., 2023; Yue et al., 2024). This conventional wisdom stems from the inherent complexity of reasoning tasks, which demand multi-step logical deduction, domain knowledge application, and structured solution paths. The resulting paradigm typically involves training on tens or hundreds of thousands of examples (Yu et al., 2024; Li et al., 2024b), based on two fundamental assumptions: first, that mastering such complex cognitive processes requires extensive supervised demonstrations, and second, that supervised fine-tuning leads primarily to memorization rather than true generalization (Zhang et al., 2024; Xu et al., 2024; Chu et al., 2025).

While this approach has shown success, it imposes substantial computational costs. We argue this data-intensive paradigm may no longer be necessary. Recent advances have transformed how LLMs acquire and utilize reasoning knowledge, suggesting a more efficient approach. Two key developments have created conditions for reimagining reasoning in LLMs: **(1) Knowledge foundation revolution**: Modern foundation models now incorporate unprecedented amounts of mathematical content during pre-training (Qwen et al., 2025; Yang et al., 2024; Wang et al., 2024). Llama 2's total training data was 1.8T tokens (Touvron et al., 2023), while Llama 3 used 3.7T tokens for mathematical reasoning (Grattafiori et al., 2024). Contemporary LLMs may already possess rich mathematical knowledge, transforming the challenge from knowledge acquisition to knowledge elicitation. **(2) Inference-time computation scaling revolution**: Techniques scaling longer reasoning chains reveal that effective reasoning requires substantial computational space during inference. Recent works (OpenAI et al., 2024; Qin et al., 2024; Huang et al., 2024; Guo et al., 2025) show that extended reasoning chains significantly improve reasoning ability. Inference-time computation provides the crucial *cognitive workspace* where models can unpack and apply their pre-trained knowledge.

We hypothesize that successful reasoning emerges from the synergy of rich pre-trained knowledge and sufficient computational resources at inference time. These developments suggest that if models possess rich reasoning knowledge and adequate computational space, activating their reasoning capabilities may require only a small number of high-quality samples that encourage extended deliberation, rather than massive fine-tuning datasets. We propose the **Less-Is-More Reasoning (LIMO) Hypothesis**, identifying two critical factors determining the *elicitation threshold* for complex reasoning: (1) the latent presence of prerequisite knowledge within the model's parameters, and (2) the effectiveness of minimal exemplars in demonstrating problem-solving processes that encourage extended deliberation. The sample efficiency of eliciting advanced reasoning is thus bounded by the model's encoded knowledge foundation and its exposure to training samples that effectively utilize inference-time computation space.

Our LIMO approach begins with a rigorous data curation process designed to identify high-quality samples that maximize reasoning elicitation. Starting from a large pool of QA pairs, we implement a multi-layered filtering system: first conducting coarse difficulty filtering to eliminate trivial problems, then fine-grained difficulty assessment to identify challenging questions, followed by knowledge point diversification to ensure comprehensive coverage. Simultaneously, we filter reasoning chains based on logical coherence, step-by-step clarity, and solution accuracy. This meticulous process yields a compact yet potent dataset of just **800** training samples. With simple supervised fine-tuning (SFT) using only this curated dataset based on Qwen2.5-32B-Instruct (Qwen et al., 2025), LIMO achieves 63.3% accuracy on the highly challenging AIME benchmark and 95.6% on MATH, outperforming previous strong SFT-based models while using just 1% of their training data. These benefits generalize across diverse previously unseen scenarios, with LIMO consistently outperforming models trained on 100x more data. This demonstrates that complex reasoning abilities can be effectively elicited through minimal but carefully curated training samples.

The main contributions of this work are: (1) We establish the LIMO hypothesis, demonstrating that complex reasoning capabilities can be elicited with just hundreds of examples by leveraging rich mathematical knowledge in pre-trained models and detailed reasoning chains. (2) Following LIMO principles, we carefully construct the LIMO dataset and fine-tune Qwen2.5-32B-Instruct through simple SFT. Our experiments demonstrate that LIMO achieves highly-competitive performance on challenging mathematical reasoning benchmarks and maintains superior out-of-distribution performance. (3) Through extensive analysis and ablation studies, we validate the effectiveness of LIMO's data selection principles and explore their applicability across different scenarios (varying foundation model knowledge, model sizes, and architectures). Additionally, we investigate LIMO's minimum data requirements to achieve competitive performance. (4) We release our models, code, and curated datasets to support future research in data-efficient reasoning.

## 2 Related Work

### 2.1 Evolution of Mathematical Reasoning in LLMs

Large-scale training data has driven reasoning abilities in LLMs. During pretraining, reasoning is enhanced by relevant corpora (Wang et al., 2024; Azerbayev et al., 2024; Paster et al., 2023; Shao et al., 2024) from textbooks, scientific papers, and mathematical code that capture diverse human cognitive patterns. Post-training research focuses on curating large-scale instruction data to teach reasoning (Yue et al., 2023; 2024; Li et al., 2024a; Yu et al., 2024) by scaling questions and solutions. While this approach has achieved significant gains, it has been criticized for relying on memorization rather than true generalization (Mirzadeh et al., 2024; Zhang et al., 2024). Mirzadeh et al. (2024) found that LLMs exhibit variance when responding to different instantiations of the same question, with performance declining when only numerical values are altered. This raises doubts about SFT methods' generalization capability (Chu et al., 2025) and whether LLMs can be true reasoners rather than merely knowledge retrievers (Kambhampati, 2024).

### 2.2 Test-time Scaling and Long Chain Reasoning

Instead of focusing on scaling model parameters and training data (Kaplan et al., 2020), recent work has shifted to exploring test-time scaling (OpenAI, 2024; Snell et al., 2024), i.e., increasing the number of tokens to improve performance. This can be achieved by augmenting LLMs with methods such as parallel sampling (Brown et al., 2024; Wang et al., 2022; Li et al., 2022) or symbolic tree search (Hao et al., 2023; Chen et al., 2024; Yao et al., 2023) to enhance reasoning ability. Furthermore, OpenAI (2024); Guo et al. (2025) explore training LLMs using reinforcement learning to generate long CoT, which often include self-reflection, verification, and backtracking—processes commonly employed by humans when solving complex problems. This approach not only innovates the training paradigm for LLMs but also provides a new form of training data to augment their reasoning ability. Our work demonstrates that this long CoT exhibits high-quality characteristics in eliciting the inherent reasoning abilities of LLMs.

### 2.3 Data Efficiency in Language Models

Zhou et al. (2023) demonstrates that with just 1,000 carefully curated prompts and responses, models can learn to follow specific formats and generalize well to unseen tasks. The findings emphasize the importance of quality over quantity in the alignment process. However, whether this lesson can be applied to reasoning tasks remains uncertain, given the potential high computational complexity of such tasks (Merrill & Sabharwal, 2024; Xiang et al., 2025). While some work on reasoning highlights the importance of quality during the curation of training data (Zhou et al., 2024; Yu et al., 2024), the quantity of such data is still much larger compared to that in LIMA. Our work extends the ideology of LIMA to reasoning tasks by investigating what constitutes high-quality questions and solutions, and demonstrates that the reasoning ability of LLMs can be enhanced in a highly data-efficient manner.

## 3 LIMO Dataset

We formalize the **Less-Is-More Reasoning (LIMO) Hypothesis** as follows: In foundation models where domain knowledge has been comprehensively encoded during pre-training, sophisticated reasoning capabilities can emerge through minimal but precisely orchestrated demonstrations of cognitive processes. This hypothesis rests on two fundamental premises: (I) The latent presence of prerequisite knowledge within the model's parameter space (II) The quality of reasoning chains that precisely decompose complex problems into detailed, logical steps, making the cognitive process explicit and traceable. To validate this hypothesis, we propose a systematic approach to construct a high-quality, minimal dataset that can effectively elicit the model's inherent reasoning capabilities.

### 3.1 High-Quality Data Curation

In this paper, we focus on reasoning tasks with verifiable answers. Given a question $q \in \mathcal{Q}$, we aim to generate an answer $a \in \mathcal{A}$ via a reasoning chain $r \in \mathcal{R}$ consisting of intermediate steps $\{s_1, s_2, ..., s_n\}$. This reasoning process is formalized as $f : \mathcal{Q} \to \mathcal{R} \times \mathcal{A}$. Dataset quality depends on both question quality and solution quality. Our curation process constructs a deliberately small, high-quality dataset $\mathcal{D} = \{(q_i, r_i, a_i)\}_{i=1}^{N}$ to validate our LIMO hypothesis that prioritizes quality over quantity.

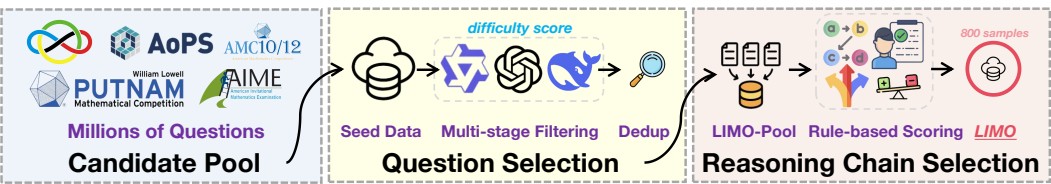

Figure 2: The LIMO dataset construction pipeline

### 3.1.1 Question Selection

We hypothesize that high-quality questions $q \in \mathcal{Q}$ should naturally elicit extended reasoning processes. Our selection criteria prioritize challenging problems that foster complex reasoning chains and knowledge integration, while also considering problems that promote exploration of diverse problem-solving approaches.

To implement these criteria effectively, we first assembled a comprehensive pool of candidate problems from various established datasets: NuminaMath-CoT (Li et al., 2024b), featuring meticulously annotated problems from high school to advanced competition levels; DeepScaleR (Luo et al., 2025), consists of approximately 40,000 unique mathematics problem-answer pairs; AIME historical examination problems before 2024, known for its extremely challenging and integrative problems spanning multiple mathematical domains; MATH (Hendrycks et al., 2021), encompassing various competitive mathematics problems from prestigious contests; and several questions including Chinese elementary, middle school, high school, and undergraduate-level exercises and examination papers.

From this extensive initial collection, we implemented a systematic multi-stage filtration pipeline (Figure 2). Starting with a corpus of tens of millions of mathematical problems, we first applied a baseline difficulty filter using a short-CoT mathematical model, Qwen2.5-Math-7B-Instruct (Yang et al., 2024). Problems that this model solved correctly within four attempts were excluded, ensuring that only non-trivial problems remained. Next, we subjected the filtered problems to a more rigorous evaluation using a stronger reasoning model, DeepSeek-R1-Distill-Qwen-32B (Guo et al., 2025). For each remaining problem, we sampled 32 solution attempts and used the empirical success rate as a difficulty indicator. Problems that were successfully solved in only 1–3 out of 32 attempts were retained, yielding a refined dataset of 2,125 problems, which composed our initial data pool (referred to as LIMO-Pool). To ensure dataset integrity, we conducted thorough deduplication against all evaluation benchmarks with n-gram matching, confirming no overlap existed.

### 3.1.2   Reasoning Chain Construction

The quality of reasoning chains fundamentally impacts the effectiveness of large language model training. To develop our solution dataset, we employed three state-of-the-art reasoning models—DeepSeek R1, DeepSeek-R1-Distill-Qwen-32B (Guo et al., 2025), and QwQ-32B (Team, 2025b)—sampling multiple solutions from each to generate diverse reasoning approaches. Subsequently, all the authors conducted a comprehensive analysis of these filtered solutions through collaborative examination. Through careful observation and systematic review, we identified several key characteristics that distinguish high-quality reasoning chains:

**Elaborated Reasoning**: Comprehensive exploration of logical steps without premature conclusions

**Self-Verification**: Regular validation of intermediate results and logical consistency

**Exploratory Approach**: Consideration of multiple possibilities before reaching conclusions

**Adaptive Granularity**: Appropriate detail level across simple and complex deductions

To quantify these qualities, we implemented a rule-based scoring system that calculated weighted metrics for each dimension. Elaborated Reasoning was measured by solution length (30% weight); Self-Verification through frequency of validation-related words like "check" and "verify" (20% weight); Exploratory Approach by counting tentative expressions such as "perhaps" and "might" (25% weight); and Adaptive Granularity via connective phrases like "therefore" and "since" (25% weight). All keyword frequencies were normalized by text length to ensure fair comparison across solutions of different sizes.

From our initial collection of 2125 questions, we selected the highest-scoring solution for each problem, ranked these pairs by quality score, and extracted the top **800** to form the LIMO Dataset. This curation process embodies our **Less-Is-More** philosophy—prioritizing demonstration quality over quantity to enhance complex reasoning capabilities.

## 4   Training Recipe

Based on the **Less-Is-More** principle, a model with substantial reasoning knowledge from pre-training and the ability to perform long-chain reasoning at test time can develop robust reasoning abilities. With exposure to just a few hundred carefully selected SFT examples, the model learns to integrate meta-reasoning tasks into cohesive reasoning chains.

We fine-tune Qwen2.5-32B-Instruct using supervised fine-tuning on our LIMO dataset. We set as our sequence length limit because all SFT response sequences remain under 16,384 tokens. The training process employs full-parameter fine-tuning with DeepSpeed ZeRO-3 optimization (Rajbhandari et al., 2020) and FlashAttention-2 (Dao, 2023). For optimization, we utilize a learning rate of 5.0e-6 with a cosine decay schedule. We deliberately omit the warmup phase to facilitate rapid adaptation to the high-quality reasoning examples in our dataset. The model is trained for 15 epochs with a batch size of 64 examples to balance computational efficiency and stable convergence.

## 5   Evaluation Framework

We establish a comprehensive evaluation framework to assess our models' mathematical reasoning capabilities across various dimensions. Our framework encompasses both in-domain and out-of-distribution evaluations, utilizing established benchmarks alongside novel multilingual tests to thoroughly examine generalization capabilities beyond the training distribution.

**In-domain Evaluation**   To comprehensively assess the models' performance across various reasoning capabilities, we have established a diverse evaluation framework encompassing both traditional and novel benchmarks. Our primary evaluation suite includes several well-established mathematical competitions and benchmarks: the American Invitational

Mathematics Examination (AIME24), MATH500 (Hendrycks et al., 2021), and the American Mathematics Competitions (AMC23).

**Out-of-distribution Evaluation**  To rigorously evaluate OOD performance, we select benchmarks differing from our training data across three categories. First, we include diverse mathematical competitions like OlympiadBench (He et al., 2024) to test performance on different mathematical challenges. Second, to minimize data contamination, we construct novel multilingual benchmarks using recent examination problems: CHMath (2024 Chinese High School Mathematics League), Gaokao (China's 2024 College Entrance Exam), Kaoyan (Chinese Graduate School Entrance Exams), and GradeSchool (our new elementary mathematics benchmark). These Chinese-language problems introduce an additional OOD dimension, assessing both cross-distribution and cross-lingual reasoning capabilities. Third, we incorporate multi-disciplinary benchmarks including Minerva (Lewkowycz et al., 2022) (undergraduate-level STEM) and GPQA (Rein et al., 2023) to evaluate the transfer of mathematical reasoning skills to broader contexts beyond our training domain.

**Performance metrics**  We evaluate performance using the pass@1 metric in a zero-shot chain-of-thought setting across all benchmarks. For larger benchmarks (MATH500, OlympiadBench, Gaokao, Kaoyan, GradeSchool, MinervaMath, and GPQA), we employ greedy decoding with a single sample. For smaller benchmarks (less than 50 problems: AIME24, AMC23, and CHMath), we generate 4 samples (temperature=0.6) and calculate the unbiased pass@1 metric (Chen et al., 2021). We use rule-based evaluations for numerical answers and an LLM-based evaluator for complex answer formats. All evaluations maintain a 32,768 token maximum output length.

# 6   Experiment

## 6.1   Baselines

We compare LIMO against a comprehensive set of baselines with several prominent models. These include **OpenAI-o1-preview** (OpenAI, 2024), a large language model that has demonstrated advanced mathematical reasoning abilities across various complex tasks; **QwQ-32B-Preview** (Team, 2024b), a model specifically designed for mathematical problem-solving with strong reasoning capabilities; and **Qwen2.5-32B-Instruct**, which serves as our base model for comparative analysis.

To investigate the impact of training data efficiency, we conduct comparative experiments using mainstream open-source reasoning datasets for supervised fine-tuning on our base model. For a fair comparison, all experiments use the same LLM backbone as LIMO, ensuring that performance differences are solely attributable to the training data characteristics. These comparative datasets include OpenThoughts-114k (Team, 2025a), a synthetic reasoning dataset containing 114k examples covering mathematics, science, coding, and puzzles, with solutions following a structured reasoning format generated by DeepSeek-R1; and **NuminaMath-100k**, a randomly selected 100k subset of NuminaMath-CoT, featuring mathematical problems ranging from Chinese high school exercises to international mathematics olympiad competitions, with each solution following a Chain of Thought (CoT) format (Wei et al., 2022).

These datasets contain substantially more samples than LIMO's training set (800 examples), allowing us to examine the relationship between data quantity and model performance.

## 6.2   Main Results

Our experimental results demonstrate LIMO's superior performance across both in-domain and out-of-domain tasks, as shown in Table 1.

**In-domain Performance**  On in-domain tasks, LIMO achieves the best results across all benchmarks. For AIME24, LIMO achieves 63.3% accuracy, outperforming QwQ-32B-Preview (50.0%) and OpenAI-o1-preview (44.6%) by significant margins. On MATH500,

| Datasets | OpenAI-o1 -preview | Qwen2.5-32B -Instruct | QwQ-32B- preview | OpenThoughts (114k) | NuminaMath (100k) | LIMO ours(800) |
|---|---|---|---|---|---|---|
| | | | In Domain | | | |
| AIME24 | 44.6 | 16.5 | 50.0 | 50.2 | 6.5 | **63.3** |
| MATH500 | 85.5 | 79.4 | 89.8 | 80.6 | 59.2 | **95.6** |
| AMC23 | 81.8 | 64.0 | 83.6 | 80.5 | 40.6 | **96.3** |
| | | | Out of Domain | | | |
| OlympiadBench | 52.1 | 45.3 | 58.5 | 56.3 | 36.7 | **67.6** |
| CHMath | 50.0 | 27.3 | 68.5 | 74.1 | 11.2 | **84.2** |
| Gaokao | 62.1 | 72.1 | 80.1 | 63.2 | 49.4 | **91.1** |
| Kaoyan | 51.5 | 48.2 | 70.3 | 54.7 | 32.7 | **83.9** |
| GradeSchool | 62.8 | 56.7 | 63.8 | 39.0 | 36.2 | **76.2** |
| Minerva | 47.1 | 41.2 | 39.0 | 41.1 | 24.6 | **52.2** |
| GPQA | **73.3** | 48.0 | 65.1 | 42.9 | 25.8 | 70.7 |
| AVG. | 61.1 | 49.9 | 66.9 | 58.3 | 32.3 | **78.1** |

Table 1: **Comparison of model performance (pass@1) across various mathematical reasoning benchmarks** Models include state-of-the-art LLMs (OpenAI-o1-preview, QwQ-32B-Preview), our base model (Qwen2.5-32B-Instruct), and models fine-tuned on different datasets. Training data sizes are shown in parentheses. Best results for each benchmark are shown in bold. Our proposed LIMO model (highlighted in blue) achieves superior performance despite using significantly fewer training examples (800) compared to other fine-tuned models (more than 100k).

LIMO achieves a notable 95.6% accuracy, surpassing QwQ-32B-Preview (89.8%) and OpenAI-o1-preview (85.5%). The performance gap is even more pronounced on AMC23, where LIMO reaches 96.3% accuracy compared to QwQ-32B-Preview's 83.6%.

**Out-of-domain Generalization** LIMO demonstrates strong generalization capabilities across diverse out-of-domain tasks. On OlympiadBench, LIMO achieves 67.6% accuracy, significantly outperforming QwQ-32B-Preview (58.5%) and the base model (45.3%). Similar improvements are observed on other challenging benchmarks such as CHMath (84.2% vs 68.5%) and GradeSchool (76.2% vs 63.8%). Notably, LIMO maintains competitive performance even on GPQA, where it achieves 70.7% accuracy, close to OpenAI-o1-preview's leading score of 73.3%.

**Comparison with Larger Datasets** Our experiments reveal that despite larger scale, both baseline datasets underperform compared to LIMO. NuminaMath-100k shows significant degradation (32.3% vs. base model's 49.9%) due to uncurated reasoning chains, while OpenThoughts-114k achieves suboptimal results (58.3%) probably due to unfocused problem selection. In contrast, LIMO's carefully curated 800 problems yield superior performance (78.1%), demonstrating that targeted selection are more crucial than data quantity for developing robust reasoning capabilities.

**Overall Performance** LIMO achieves the highest average performance of 78.1% across all benchmarks, substantially outperforming OpenAI-o1-preview, QwQ-32B-Preview, and other baselines. This comprehensive evaluation demonstrates that LIMO's carefully curated training approach with just 800 examples can outperform models trained on datasets that are orders of magnitude larger.

## 6.3 Analysis

### 6.3.1 RQ1: Impact of Reasoning Chain Quality

To further validate and compare the effectiveness of our LIMO method, we first focus on the selection of high-quality reasoning chains. We investigate what characteristics define superior reasoning chains that lead to better model performance through a controlled comparative study of solutions with varying quality for identical problems.

For this analysis, we select 500 questions from the LIMO dataset that each have multiple correct solutions generated by diverse models, ensuring we have varied yet accurate approaches to the same problems. We collect and categorize these solutions into five distinct quality levels (L1-L5, with L5 being the highest) based on our rule-based scoring system described in Section 3.1.2.

Results in Figure 3 show clear correlation between reasoning quality and model performance. L5-trained models achieve highest results on both AIME24 and MATH500, with performance decreasing consistently with each quality level. The substantial gap between L5 and L1 solutions demonstrates that reasoning chain quality significantly influences model performance, underscoring the importance of curating high-quality training data.

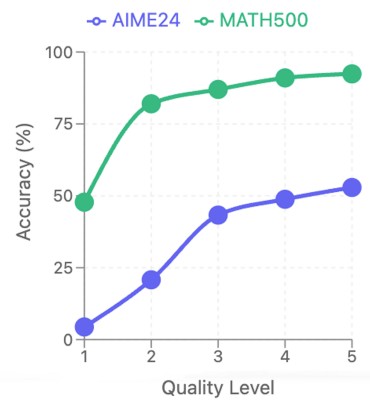

Figure 3: Comparison of models trained on reasoning chains of different quality levels.

### 6.3.2 RQ2: Impact of Question Quality

We hypothesize that more challenging problems foster complex reasoning chains and enhanced knowledge integration. To test this, we examine how question difficulty affects models' reasoning capabilities. We select three sets of 500 problems with increasing difficulty: **Simple-500** (MATH levels 1-2), **Complex-500** (MATH levels 3-5), and **Advanced-500** (AIME problems). We verify the difficulty gradient by evaluating various LLMs, observing declining accuracy and increasing solution length across these sets. We then use DeepSeek-R1 to generate high-quality solutions for each set and fine-tune Qwen2.5-32B-Instruct on them. Results in Figure 4 show that modifying problem selection alone leads to 16% accuracy improvement on AIME2024, reaching 51.5%. Notably, the model fine-tuned on Advanced-500 achieves 91.2% on MATH500 despite no in-domain training data, suggesting that reasoning improvements from increased problem difficulty generalize across datasets.

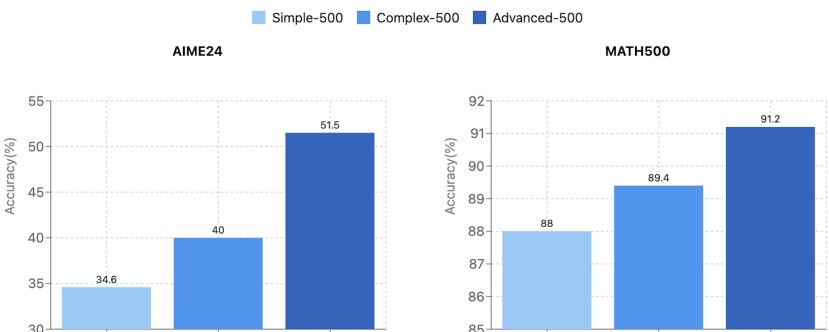

Figure 4: Performance comparison on MATH and AIME benchmarks between models trained on different question quality: Simple-500, Complex-500, and Advanced-500.

### 6.3.3 RQ3: LLM Backbone - Pre-trained Knowledge

Building on our LIMO hypothesis that emphasizes latent prerequisite knowledge within model parameters, we examine how pre-training data affects a model's ability to leverage minimal exemplars for math reasoning. To isolate pre-training impact while controlling for architecture and fine-tuning, we compare two 32B-parameter models: Qwen1.5-32B-Chat (Team, 2024a) and Qwen2.5-32B-Instruct (LIMO's base model). While sharing identical architecture, Qwen2.5 features improved pre-training data quality, particularly in mathematical and code-related content. We SFT both models using identical LIMO datasets and evaluate on AIME2024 and MATH500 benchmarks.

Results in Figure 5 show pre-trained model choice dramatically impacts reasoning performance. LIMO (built on Qwen2.5) achieves 63.3% accuracy on AIME2024, a 54.1 percentage point improvement over Qwen1.5's 9.2%. On MATH500, LIMO reaches 95.6% accuracy, surpassing Qwen1.5 by 30.4 percentage points. These substantial improvements confirm that Qwen2.5's enhanced pre-training creates a stronger foundation for mathematical reasoning, aligning with our hypothesis that richer pre-trained knowledge enables more effective utilization of minimal exemplars.

It is hypothesized that increasing the number of parameters in a large language model enhances its capacity for deep reasoning and overall performance in complex tasks. To investigate this hypothesis, we fine-tune models from the Qwen2.5-Instruct series of varying sizes—3B, 7B, 14B, 32B, and 72B—using the LIMO dataset with 800 high-quality samples. The models are subjected to the same supervised fine-tuning recipe to ensure that any observed differences in performance are primarily attributable to model size.

**Qwen1.5-32B-Chat**

9.2%
AIME2024

65.2%
MATH500

**Qwen2.5-32B-Instruct**

63.3%
AIME2024

95.6%
MATH500

Figure 5: Impact of pre-trained model choice on mathematical reasoning performance.

### 6.3.4   RQ4: LLM Backbone - Model Size

Figure 6 summarizes the performance of the models on both benchmarks. The performance on AIME24 shows a marked increase with model size, rising from 2.5 for the 3B model to 68.3 for the 72B model. This supports the hypothesis that larger models are better able to handle the deep reasoning required by competition-level math problems. While the improvements on MATH500 are also evident, the gains are less dramatic, suggesting that even smaller models can achieve high accuracy on easier benchmarks like MATH500. The difference in performance between the 32B and 72B models is marginal on MATH500 (95.6% vs. 94.8%) and slightly less pronounced on AIME24 (63.3% vs. 68.3%). This indicates a potential saturation point, where increasing the number of parameters further may yield diminishing returns in fine-tuning performance for these benchmarks.

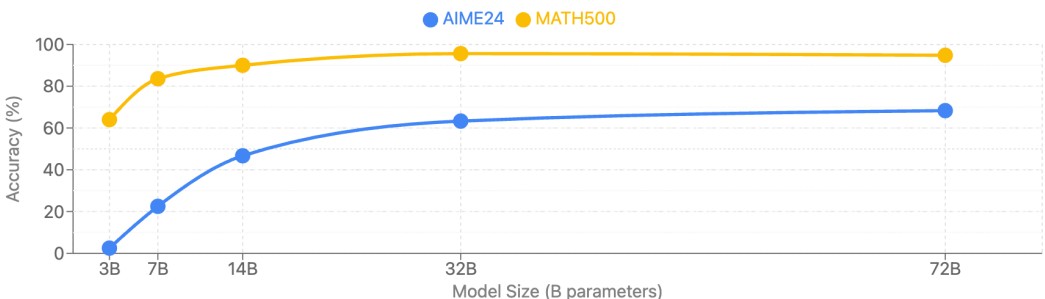

Figure 6: Scaling of mathematical reasoning ability with model size

### 6.3.5   RQ5: Sample Efficiency

Our experiments reveal that a surprisingly small number (i.e. 800) of samples can elicit competition-level mathematical reasoning, though the lower bound for maintaining effective performance remains an open question. We explore the impact of dataset size on fine-tuning efficacy by systematically varying the number of training samples. To optimize performance with constrained dataset sizes, we rank all 2,125 questions in the LIMO-pool dataset based on their best reasoning chains' quality scores. From this ranked set, we select top-ranked subsets of varying sizes—400, 800, 1,200, 1,600, and 2,000 questions—to construct five datasets (LIMO-400 through LIMO-2k). Each dataset is used to fine-tune our base model following an identical training recipe to ensure comparability.

Figure 7 and Figure 8 present the performance of models fine-tuned on different dataset sizes. Several key observations emerge: (1) Fine-tuning with just 400 samples yields dramatic improvement over the base model, increasing AIME24 accuracy from 16.5 to 57.5 and MATH500 accuracy from 79.4 to 94.8. (2) While performance continues to improve with larger datasets, we observe diminishing returns beyond 800 samples, with only marginal increases between LIMO-800 and LIMO-1k2 (+0.9 on AIME24, -0.2 on MATH500), suggesting an early plateau effect. (3) The highest dataset size (2k) provides the best overall performance (69.6 on AIME24, 95.8 on MATH500), but with minimal improvement over 1.6k, indicating that beyond a certain threshold, additional data contributes less significantly to fine-tuning gains. These findings highlight the efficiency of high-quality, carefully selected data in enhancing LLM mathematical reasoning. Future work could explore active learning strategies to further optimize sample efficiency.

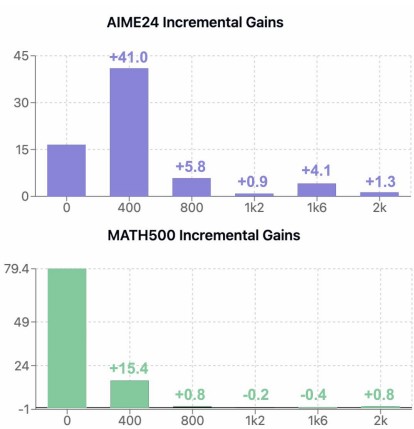

Figure 7: Incremental performance gains by dataset size.

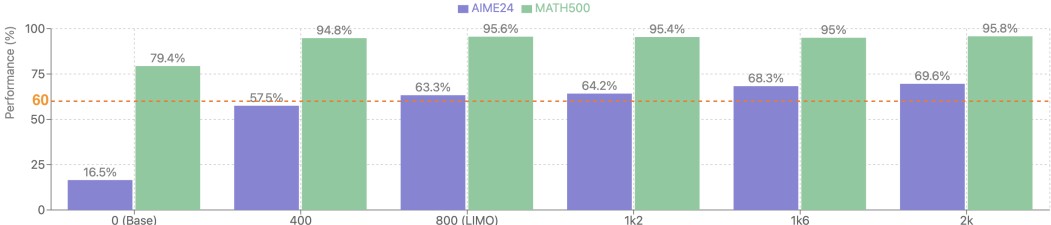

Figure 8: Impact of dataset size on model performance

# 7 Conclusion

Our work demonstrates that complex mathematical reasoning in LLMs can be achieved with surprisingly few examples, contradicting the prevailing assumption that massive training data is necessary. LIMO achieves competitive performance on challenging benchmarks using only 1% of the training data required by previous approaches. This confirms our Less-Is-More Reasoning Hypothesis: in knowledge-rich foundation models, sophisticated reasoning emerges through minimal but precisely orchestrated demonstrations that effectively utilize inference-time computation.

# 8 Acknowledgement

We would like to express our sincere gratitude to Yixiu Liu and Yiwei Qin for their valuable contributions to this research work. Their expertise, dedication, and collaborative spirit have significantly enhanced the quality of our study. Their insightful suggestions and technical assistance were instrumental in achieving our research objectives. We are also grateful to Run-Ze Fan for his diligent efforts in labeling our internal benchmark. We also wish to extend our appreciation to Haoyang Zou and Xuefeng Li for their valuable discussions during the early stages of this work. Their perspectives and insights helped shape the foundation of our research.

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
