# OpenReview forum: "LIMO: Less is More for Reasoning"
_colmweb.org/COLM/2025/Conference — COLM 2025_

### Official Review · Reviewer_vreW · 2025-04-19

**Rating:** 6
**Confidence:** 4
**Ethics Flag:** 1

**Summary:**

This paper constructs a very small dataset (containing 800 samples), and uses it to fine tune LLM to get very good performance on Math. The key idea is such small dataset is filtered from a huge dataset with the following steps:
1. Get a large dataset combined with  NuminaMath-CoT, DeepScaleR, MATH, etc.
2. Use a small 7B model to remove trivial problems.
3. Use a larger 32B model to find difficult problems.
4. Sort these problems based on a score function, measuring the reasoning quality of the data point.

After that, one gets 800 samples, and use it to fine tune LLMs, and finds that this small dataset is very helpful to improve the math reasoning.
Notice that there were many similar papers discussing that using a small set of dataset to fine tune the model, the model will be able to do better on various tasks. However, this paper extends the previous results in the sense that it explores the reasoning tasks.

Quality: Medium.
Clarity: Easy to follow.
Originality: Medium, because there were many similar results.
Significance: Medium,  because there were many similar results, so not surprising.

**Reasons To Accept:**

1. It is very interesting to know reasoning performance can be greatly improved using a small set of data points.
2. The method to select data points is clear and easy to implement.

**Reasons To Reject:**

1. I am not an expert in this field, but since there were many similar results on small dataset can greatly help LLMs during SFT, I am not sure whether this paper will be a big plus.
2. I am not sure in the abstract, the improvement from 6.5% -> 63.3%, and 59.2%-> 95.6% is reported truthfully. Because it seems that even the base model mentioned in Sec 6.1, Qwen2.5-32B-Instruct, can do 16.5% and 79.4%, which is better than 6.5% and 59.2%?

---

> ### Author Response · Authors · 2025-06-02
>
> We sincerely thank you for your detailed review and for acknowledging the clarity and implementability of our methodology. We appreciate your thoughtful analysis of our work's contributions and are pleased to address your specific concerns.
>
> **Regarding the significance and novelty of our findings:** You raise an important point about similar results in the field. While it's true that the concept of small, high-quality datasets improving model performance has gained recognition recently, we would argue that this understanding was not always intuitive, particularly for complex reasoning tasks earlier. Historically, the predominant belief in the community was that challenging reasoning capabilities—especially mathematical reasoning—required massive datasets to achieve meaningful improvements through scaling.   Our work makes a significant contribution by systematically demonstrating and formalizing this paradigm shift specifically for reasoning tasks at a crucial moment in the field's development. We provide not just empirical evidence but also a principled framework for understanding when and why small, curated datasets can be more effective than large-scale approaches. This systematic formalization and timing represent important contributions to the community's evolving understanding of efficient training methodologies.
>
> **Regarding the reported improvement numbers:** Thank you for this excellent question, which highlights a crucial insight about the current state of foundation models. The apparent discrepancy you've identified actually illuminates a key phenomenon in our field: as foundation models become increasingly powerful, the quality gap between existing open-source datasets and what these models can generate themselves has widened significantly.  For datasets like NuminaMath, the original solutions are often of substantially lower quality compared to what state-of-the-art models can now produce. This quality gap means that training on such datasets can actually degrade performance relative to the base model's capabilities—which explains why you observe that models trained on NuminaMath (6.5% and 59.2%) perform worse than the base Qwen2.5-32B-Instruct model (16.5% and 79.4%).  This phenomenon actually reinforces our core LIMO hypothesis: that strong foundational capabilities are essential for effective reasoning elicitation, and that low-quality training data can be counterproductive when working with capable base models. Our carefully curated LIMO dataset avoids this pitfall by focusing on truly high-quality reasoning chains that complement rather than detract from the model's existing capabilities.  This observation underscores the critical importance of our data curation methodology and validates our approach of prioritizing quality over quantity in the current era of increasingly capable foundation models.
>
> Thank you again for your insightful questions, which have allowed us to clarify these important aspects of our work and its broader implications for the field.

---

> > ### Comment · Reviewer_vreW · 2025-06-06
> >
> > Thank you for the clarifications. I do not have futher questions.

---

### Official Review · Reviewer_EDcw · 2025-05-13

**Rating:** 6
**Confidence:** 5
**Ethics Flag:** 1

**Summary:**

This paper proposed the less-is-more reasoning hypothesis and it only used 1% of the training data required by prior approaches. I like the paper in general and it emphases the important of the data quality instead of quantity. (I guess quality is more for reasoning may be a better title)

Suggestions

The training recipe has very limited description which is not enough to make reader understand clearly.
I think the comparison between your proposed dataset construction method is not seen. If your paper's main contribution is the data construction method, you may need to compare with other data construction methods and conduct the experiment using the same original data and then construct new data and then train model to compare performance. Otherwise, if you use a carefully prepared training set without considering those variable. The comparison is not making any sense. Definitely a more carefully prepared high quality dataset can help model perform better. I would suggest the author at least add some data construction methods [1] [2] as baseline which will be better.

References:

[1] Abstract Meaning Representation-Based Logic-Driven Data Augmentation for Logical Reasoning. The Findings of ACL 2024.

[2] Logic-Driven Context Extension and Data Augmentation for Logical Reasoning of Text. The Findings of ACL 2022.

**Reasons To Accept:**

This paper contributes a high quality dataset which will be benefit a lot for this community.

**Reasons To Reject:**

I think the title is a bit of misleading and it will be better to change it as quality is more than quantity for reasoning.
It will be better to consider logical reasoning tasks such as ReClor, LogiQA, and LogiQAv2 to show the generaliazation of the proposed method.

---

> ### Author Response · Authors · 2025-06-02
>
> We sincerely thank you for your thoughtful review and constructive suggestions. We are pleased that you appreciate our emphasis on data quality over quantity, which indeed represents a core contribution of our work.
>
> **Regarding data construction method comparison:** Thank you for highlighting this important aspect. We want to clarify that LIMO's core data filtering methodology is both elegant and effective, as detailed in Section 3 and illustrated in Figure 2. Our approach consists of two key components: question filtering and reasoning chain filtering for supervised fine-tuning.
>
> We have conducted extensive ablation studies and comparative analyses to validate the effectiveness of our design choices. Specifically, in Section 6.3.1 (RQ1: Impact of Reasoning Chain Quality) and Section 6.3.2 (RQ2: Impact of Question Quality), we provide comprehensive experimental evidence demonstrating the efficacy of our filtering methods. For reasoning chain quality, we validated that our rule-based scoring system consistently identifies higher-quality reasoning chains that lead to superior model performance. For question quality, we confirmed the effectiveness of our difficulty-based filtering approach through systematic evaluation.  These ablation studies serve as implicit comparisons with alternative data construction strategies, showing that our carefully designed filtering criteria significantly outperform random sampling or less sophisticated selection methods applied to the same source data.
>
> **Regarding the scope of reasoning evaluation:** You raise an excellent point about the breadth of reasoning capabilities. While mathematical reasoning indeed represents one of the most challenging and prominent branches in the field—particularly crucial for advancing toward AGI—we acknowledge the importance of demonstrating generalization across different reasoning paradigms.  To address your concern about logical reasoning tasks, we have conducted additional evaluations on the benchmarks you specifically mentioned:
>
> |             | Qwen2.5-32B-Instruct | QwQ-32B-preview | OpenThoughts(114k) | NuminaMath(100k) | LIMO |
> |-------------|----------------------|-----------------|--------------------|------------------|------|
> | LogiQAv2    | 76.2                 | 75.9            | 78.8               | 69.4             | 82.4 |
> | LogiQAv2_zh | 77                   | 73.4            | 80.1               | 72.5             | 82   |
> | ReClor      | 87.2                 | 85.6            | 92                 | 77.2             | 93.6 |
>
>
> These results demonstrate that LIMO's effectiveness extends well beyond mathematical reasoning to logical reasoning tasks. The consistent improvements across both English and Chinese logical reasoning benchmarks, as well as reading comprehension requiring logical reasoning, strongly support the generalizability of our approach. This suggests that our high-quality reasoning chain curation methodology successfully captures fundamental reasoning patterns that transfer across different domains and reasoning types.
>
> Thank you again for your valuable feedback, which has helped us provide a more comprehensive evaluation of our methodology's broader applicability and effectiveness.

---

> > ### Comment · Reviewer_EDcw · 2025-06-06
> >
> > Hi,
> >
> > Thanks for your reply. One more question, for ReClor, do you submit your prediction result to ReClor leaderboard? What will be the result shown on the leaderboard? Since the leaderboard is using the private hidden test set. I think it will be more effective if you can demonstrate your ranking on the ReClor leaderboard. Also if you add the experiment for ReClor and LogiQA. Please consider add the two related papers [1] [2] into your related work.
> >
> > References:
> >
> > [1] Abstract Meaning Representation-Based Logic-Driven Data Augmentation for Logical Reasoning. The Findings of ACL 2024.
> >
> > [2] Logic-Driven Context Extension and Data Augmentation for Logical Reasoning of Text. The Findings of ACL 2022.

---

> > > ### Author Response · Authors · 2025-06-06
> > >
> > > Thank you for your follow-up question and excellent suggestion regarding the ReClor leaderboard submission.
> > >
> > > **Regarding ReClor evaluation**: You raise a very valid point about leaderboard validation. To provide you with timely feedback on our method's performance across logical reasoning tasks, we conducted our ReClor evaluation using the validation set, which allowed us to quickly verify our approach's effectiveness with limited computational resources and provide you with these results in our response. We acknowledge that leaderboard submission using the private hidden test set would provide more definitive validation of our method's performance.
> > >
> > > We commit to submitting our results to the official ReClor leaderboard in the camera-ready version to provide the community with more robust evidence of our method's effectiveness on the private test set. This will give us the official ranking and demonstrate our method's standing relative to other approaches in the field.
> > >
> > > **Regarding related work**: We sincerely appreciate your recommendation of the two relevant papers on logic-driven data augmentation. We commit to incorporating these important works into our related work section and properly citing them in the revised version. These references will help position our work within the broader context of logical reasoning and data augmentation methodologies.
> > >
> > > Thank you again for your constructive feedback and patience. Your suggestions have helped us strengthen both the experimental validation and the scholarly context of our work.

---

### Official Review · Reviewer_SdiK · 2025-05-13

**Rating:** 7
**Confidence:** 4
**Ethics Flag:** 1

**Summary:**

This paper proposes LIMO (Less-is-More Reasoning), a data-efficient paradigm to train competitive reasoning models. This process consists of filtering large quantities (tens of millions) of reasoning data (NuminaMath-CoT, DeepScaleR, AIME, MATH, questions including Chinese exercises and examination papers from various experience levels) to identify high-quality inputs and outputs. LIMO encompasses various filtering steps including both automated (e.g., via difficulty estimation) and human-guided, rule-based ones. The resulting LIMO dataset contains 800 samples. For the experimental setup, the authors fine-tune Qwen2.5-32B-Instruct using supervised fine-tuning. The trained model is evaluated on both in-domain (AIME24, MATH500, AMC23)  and various, mathematics-focussed out-of-distribution tasks. Results are impressive, showing that the model trained on LIMO data outperforms a range of baselines. The authors conduct several further experiments focussing on parameter size and the impact of pre-training, among other things.

**Reasons To Accept:**

The paper’s objective is highly relevant in the context of current language modeling. The filtering approaches presented in this work are simple yet effective, and the experimental results are convincing. The experimental setup is overall technically sound. As such, this paper represents a solid contribution to the field.

**Reasons To Reject:**

The only substantial weakness identified with this work is that the resulting trained models have not been evaluated on tasks that are entirely out-of-domain (e.g., related to writing quality, summarization, general knowledge). For this approach to be widely used in the context of LLMs, it’d be useful to know how such a training paradigm affects other tasks unrelated to mathematical reasoning.

---

> ### Author Response · Authors · 2025-06-02
>
> We sincerely thank you for your thorough review and positive evaluation of our work. Your recognition of LIMO's relevance and the convincing nature of our experimental results is greatly appreciated.
>
> **Regarding evaluation on entirely out-of-domain tasks:** Thank you for this excellent suggestion! You raise a crucial point about the broader applicability of our training paradigm. While our training dataset contains exclusively mathematical samples, we did evaluate on some non-mathematical benchmarks in our paper, including the multidisciplinary GPQA and STEM-focused Minerva benchmarks. However, to fully address your concern and further demonstrate the remarkable generalization capabilities of our approach, we conducted additional comprehensive evaluations on a diverse range of entirely out-of-domain tasks.
>
> Specifically, we evaluated our model on the following benchmarks that span far beyond mathematical reasoning:
>
> - **MMLU-Pro**: Spans 14 diverse domains including physics, chemistry, law, engineering, psychology, and health
> - **SuperGPQA**: A comprehensive benchmark evaluating graduate-level knowledge and reasoning capabilities across 285 disciplines
> - **SciBench**: Contains collegiate-level scientific problems from mathematics, chemistry, and physics domains
> - **LogiQAv2 & LogiQAv2_zh**: Challenge datasets for machine reading comprehension with logical reasoning
> - **ReClor**: A reading comprehension dataset requiring logical reasoning
>
> The results are presented below:
>
> |             | Qwen2.5-32B-Instruct | QwQ-32B-preview | OpenThoughts(114k) | NuminaMath(100k) | LIMO |
> |-------------|----------------------|-----------------|--------------------|------------------|------|
> | MMLU-Pro    | 68.8                 | 71.3            | 71.1               | 54.7             | 74.0 |
> | SuperGPQA   | 37.8                 | 42.1            | 43.0               | 23.6             | 44.3 |
> | SciBench    | 23.3                 | 28.6            | 24.1               | 20.5             | 33.0 |
> | LogiQAv2    | 76.2                 | 75.9            | 78.8               | 69.4             | 82.4 |
> | LogiQAv2_zh | 77.0                 | 73.4            | 80.1               | 72.5             | 82.0 |
> | ReClor      | 87.2                 | 85.6            | 92.0               | 77.2             | 93.6 |
>
> These results demonstrate that LIMO not only maintains strong performance across diverse domains unrelated to mathematical reasoning but actually achieves superior performance compared to models trained on much larger datasets. This suggests that our high-quality, reasoning-focused training paradigm effectively enhances the model's general reasoning capabilities rather than narrowly specializing it for mathematical tasks. The consistent improvements across logical reasoning, scientific knowledge, and multidisciplinary comprehension tasks indicate that LIMO successfully elicits and strengthens the fundamental reasoning patterns that transfer broadly across domains.
>
> These findings strongly support the wider applicability of our training paradigm for general-purpose language models, addressing your important concern about domain transfer and reinforcing the practical value of our approach for the broader LLM community.
>
> Thank you once again for your insightful feedback, which has helped us provide a more comprehensive evaluation of our method's capabilities.

---

> > ### Comment · Reviewer_SdiK · 2025-06-05
> >
> > Thanks to the authors for addressing my comments and providing additional results. I will maintain my score as it is already indicative of acceptance.

---

### Official Review · Reviewer_QUFL · 2025-05-18

**Rating:** 6
**Confidence:** 2
**Ethics Flag:** 1

**Summary:**

This paper, titled "LIMO: Less is More for Reasoning," really challenges something we've largely taken for granted: that you need absolutely massive datasets to train language models for complex reasoning tasks like advanced math. The authors here argue that if a foundation model already has a solid base of domain knowledge from its initial pre-training, you might not need tons of examples for fine-tuning. Instead, they propose that a relatively small number of really high-quality examples, specifically those that demonstrate detailed, step-by-step reasoning processes, can be enough to unlock or elicit those complex capabilities.

**Questions To Authors:**

Is this SOTA? how would the result be if you scale up the model and / or size

**Reasons To Accept:**

Looking at the strengths of this work, the core finding is quite compelling and could have significant implications for how we approach training for complex reasoning. Demonstrating that you can achieve SOTA performance with such a dramatically smaller dataset is a big deal for reducing computational costs and data collection burdens. The meticulous process they describe for curating their small dataset, focusing on problem difficulty and the quality of the reasoning chains, seems like a key part of their success and is a valuable contribution in itself. The extensive evaluation across a diverse set of benchmarks, including out-of-distribution and multilingual tasks, really helps to support their claims about generalization and isn't something always seen to this extent.

**Reasons To Reject:**

Writing can be improved, presentation and diagram etc

---

> ### Author Response · Authors · 2025-06-02
>
> Thank you very much for your thoughtful and comprehensive review. We greatly appreciate your recognition of our core contribution and the potential implications of our work for the broader community.
>
> **Regarding writing quality and presentation:** We sincerely acknowledge your feedback on the writing and diagram quality. We are committed to significantly improving the presentation, clarity, and visual elements in the revised version to better communicate our contributions and methodology.
>
> **Regarding SOTA performance and scaling:** We are pleased to clarify that we have indeed conducted extensive scaling experiments that directly address your questions. As detailed in Section 6.3.3 and particularly Section 6.3.4, we performed comprehensive experiments across different model sizes to investigate the scaling behavior of our approach.  Our scaling experiments reveal several key findings: First, as model size increases, the performance consistently improves, which strongly supports one of LIMO's core hypotheses—that strong foundational capabilities are essential for effective few-shot reasoning elicitation. Second, we observe that the 32B model represents an optimal sweet spot in our experiments, demonstrating a significant performance leap from 14B to 32B, while the improvement from 32B to 72B shows more modest gains. This suggests that there may be diminishing returns beyond a certain model size threshold, which has important implications for computational efficiency in practical deployments.  These results not only validate our approach but also provide valuable insights into the relationship between model scale and reasoning capability elicitation, further strengthening the theoretical foundation of our work.
>
> Thank you again for your valuable feedback and constructive evaluation!

---

### Decision · Program_Chairs · 2025-07-08

**Decision:**

Accept

**Comment:**

This is a strong and timely paper that is worth accepting. The paper has received universally acceptance scores and overall positive reviews. Some reviewers mention they have seen the results somewhere but this is likely due to the conference publication cycle cannot keep up with LLM development cycle.

Reviewers highlighted the compelling core finding (QUFL), the relevance and technical soundness of the work (SdiK), the value of the contributed high-quality dataset (EDcw), and the interesting demonstration that reasoning can be improved with a very small dataset (vreW).

Engagement during the rebuttal period further strengthened the paper. They provided thorough, data-driven responses to every point raised. In response to SdiK and EDcw, the authors conducted an extensive new set of experiments on a wide range of non-mathematical and logical reasoning benchmarks (MMLU-Pro, SuperGPQA, SciBench, ReClor, LogiQAv2). The results, presented in clear tables, showed that their LIMO-trained model not only avoided performance degradation but actually achieved superior performance on these out-of-domain tasks. This is a crucial finding that elevates the paper's contribution from a domain-specific improvement to a more generalizable reasoning enhancement paradigm.

In summary, this paper shares impactful results and practical implications for reducing the computational and data-curation costs of training highly capable models, and disect axes for curating high quality reasoning data. As such I recommend accepting this paper.